# Characterization of New Proteomic Biomarker Candidates in Mucopolysaccharidosis Type IVA

**DOI:** 10.3390/ijms22010226

**Published:** 2020-12-28

**Authors:** Víctor J. Álvarez, Susana B. Bravo, Maria Pilar Chantada-Vazquez, Cristóbal Colón, María J. De Castro, Montserrat Morales, Isidro Vitoria, Shunji Tomatsu, Francisco J. Otero-Espinar, María L. Couce

**Affiliations:** 1Department of Forensic Sciences, Pathology, Gynecology and Obstetrics, Pediatrics, Neonatology Service, Department of Paediatrics, Hospital Clínico Universitario de Santiago de Compostela, Health Research Institute of Santiago de Compostela (IDIS), CIBERER, MetabERN, 15706 Santiago de Compostela, Spain or jvictor_alvarez@hotmail.com (V.J.Á.); cristobal.colon.mejeras@sergas.es (C.C.); mj.decastrol@gmail.com (M.J.D.C.); 2Skeletal Dysplasia Lab Nemours Biomedical Research Nemours/Alfred I. du Pont Hospital for Children, 1600 Rockland Road., Wilmington, DE 19803, USA; shunji.tomatsu@nemours.org; 3Proteomic Platform, Health Research Institute of Santiago de Compostela (IDIS), Hospital Clínico Universitario de Santiago de Compostela, 15706 Santiago de Compostela, Spain; sbbravo@gmail.com (S.B.B.); mariadelpilarchantadavazquez@gmail.com (M.P.C.-V.); 4Minority Diseases Unit Hospital Universitario12 de Octubre, 28041 Madrid, Spain; moralmon@yahoo.es; 5Nutrition and Metabolophaties Unit, Hospital Universitario La Fe, 46026 Valencia, Spain; vitoria_isi@gva.es; 6Paraquasil Platform, Health Research Institute of Santiago de Compostela (IDIS), Hospital Clínico Universitario de Santiago de Compostela, 15706 Santiago de Compostela, Spain; francisco.otero@usc.es; 7Department of Pharmacology, Pharmacy and Pharmaceutical Technology, School of Pharmacy, Campus Vida, University of Santiago de Compostela, 15872 Santiago de Compostela, Spain

**Keywords:** biomarkers, enzyme replacement therapy, lysosomal disorders, proteomics

## Abstract

Mucopolysaccharidosis type IVA (MPS IVA) is a lysosomal storage disease caused by mutations in the *N*-acetylgalactosamine-6-sulfatase (*GALNS*) gene. Skeletal dysplasia and the related clinical features of MPS IVA are caused by disruption of the cartilage and its extracellular matrix, leading to a growth imbalance. Enzyme replacement therapy (ERT) with recombinant human GALNS has yielded positive results in activity of daily living and endurance tests. However, no data have demonstrated improvements in bone lesions and bone grow thin MPS IVA after ERT, and there is no correlation between therapeutic efficacy and urine levels of keratan sulfate, which accumulates in MPS IVA patients. Using qualitative and quantitative proteomics approaches, we analyzed leukocyte samples from healthy controls (*n* = 6) and from untreated (*n* = 5) and ERT-treated (*n* = 8, sampled before and after treatment) MPS IVA patients to identify potential biomarkers of disease. Out of 690 proteins identified in leukocytes, we selected a group of proteins that were dysregulated in MPS IVA patients with ERT. From these, we identified four potential protein biomarkers, all of which may influence bone and cartilage metabolism: lactotransferrin, coronin 1A, neutral alpha-glucosidase AB, and vitronectin. Further studies of cartilage and bone alterations in MPS IVA will be required to verify the validity of these proteins as potential biomarkers of MPS IVA.

## 1. Introduction

Morquio A syndrome, or mucopolysaccharidosis type IVA (MPS IVA, OMIM #253000), is an autosomal recessive disease caused by mutations in the *GALNS* gene. A deficiency of *N*-acetylgalactosamine-6-sulfatase (GALNS, E.C.3.1.6.4) [1,2,3] leads to the accumulation of keratan sulfate (KS) and chondroitin-6-sulfate (C6S) in multiple tissues, mainly bone, cartilage, heart valves, and cornea. The classical phenotype is characterized by systemic skeletal dysplasia with incomplete ossification and successive imbalance of growth [4], including short stature and neck, cervical instability, spinal cord compression, tracheal obstruction, prominent chest, kyphoscoliosis, laxity of joints, hip dysplasia, and knock knees [5,6]. Respiratory failure is the primary cause of death during the second and third decades of life in untreated patients [7,8].

Currently, two therapies are available for MPS IVA in clinical practice, namely enzyme replacement therapy (ERT) and hematopoietic stem cell transplantation (HSCT) [1,9,10]. ERT and HSCT are based on the principle of cross-correction, whereby lysosomal enzymes are taken up by deficient recipients’ cells and their lysosomes via the mannose-6-phosphate receptor. ERT with recombinant enzyme GALNS (elosulfase alfa) is an established treatment for MPS IVA. However, like other forms of ERT for lysosomal storage diseases (LSDs), elosulfase alfa has several limitations. It requires weekly 4–6 h infusions; is cleared rapidly due to its short half-life (35 min in human, 2 min in mouse) [11,12,13], it is expensive [13,14], penetration of the avascular cartilage is limited; and patients can develop an immune response against the infused enzyme [11,14,15]. In addition, clinical trials of elosulfase alfa have shown little improvement in bone growth and pathology [16,17,18,19]. Recent studies have demonstrated that the presence of KS, KS sulfation level, chondroitin-6-sulfate levels, and the presence of collagen type II in blood are potential biomarkers [20] associated with bone and cartilage disease in MPS IVA. Although urinary KS levels have been measured as a potential biomarker in clinical trials, there is no proof that decreases in urinary KS reduction correlate with clinical improvement. Urinary KS originates in the kidneys and does not reflect the degree of impairment of bone and other relevant tissues in MPS IVA. Therefore, urinary KS is considered a pharmacokinetic marker but not a surrogate biomarker [20].

Proteomics allows the large-scale identification and quantification of proteins in biological fluids (serum, urine, saliva, tears, etc.), cells, tissues, or organisms in continuous change. Among the first proteomic approaches developed are one and two-dimensional polyacrylamide gel systems (1D SDS-PAGE and 2D SDS-PAGE). These methods provide a means of monitoring protein purification, protein expression, and post-transcriptional modifications [20]. Although 1D and 2D SDS-PAGE are commonly used to screen for putative biomarkers in several disorders, particularly in animal studies [21,22], they have not been applied to LSDs. Mass spectrometry (MS) is a more recent and accurate proteomics technology, and allows high-through put protein identification [23,24]. This approach also enables the characterization of post-transcriptional modifications, such as phosphorylation and glycosylation, similar to the 2D-PAGE approach [25]. In the last decade, MS has led to significant advances by increasing the number of potential protein biomarkers of different diseases [26]. Liquid chromatography (LC) coupled with MS is widely used to search for disease biomarkers [24] in different biological sample types [26,27,28].

Most quantitative methods using liquid chromatography-tandem mass spectrometry (LC-MS/MS) involve labeling of proteins/peptides with heavy and light stable-isotope pairs (SILAC, iTRAQ). More recently developed label-free quantification techniques rely on advanced software analysis. One of these techniques is SWATH-MS (sequential window acquisition of all theoretical mass spectra) analysis [29,30]. This method measures the concentrations of peptide analytes (10 peptides per protein in SWATH analysis) in two or more samples, using certain peptides in the samples as an internal standard. By contrast, other absolute quantification techniques use external peptides as standards to create a calibration curve. These external peptides are identified by searching using monitoring methods (monitoring of selected reactions or multiple reactions) [23].

Given that MPS IVA is a dynamic entity that involves alterations in the expression of multiple proteins, proteomics techniques can provide important biological information that can help advance our understanding of the underlying pathophysiological mechanisms. Moreover, proteomics could aid the identification of novel disease biomarkers [28,30,31], which can facilitate disease diagnosis, prognosis, and monitoring [32,33,34,35,36,37,38,39,40,41,42].

To address the lack of disease-specific protein biomarkers in MPS IVA, we applied quantitative and qualitative proteomics approaches to peripheral blood cell specimens from treated and untreated patients with MPS IVA to systematically identify and biochemically characterize new biomarker candidates.

## 2. Results

Leukocytesamplesfrom13 patients with MPS IVA and 6 control subjects were classified in the three following groups: healthy controls; untreated MPS IVA patients; and ERT-treated MPS IVA patients, from whom samples were acquired before (ERT-a group) and 24 h after (ERT-b group) treatment. The demographic features of participating MPS IVA patients are shown in Table 1.

We first performed a qualitative analysis to identify the entire set of proteins expressed in leukocyte samples using LC-MS/MS technology in data-dependent acquisition (DDA) mode. Proteins were identified with only 1% error (false discovery rate (FDR) 1%). To characterize the most significant proteins, we selected those commonly found in all or all but one sample (*n* − 1) per group. Subsequently, quantitative analysis was performed using the SWATH method.

### 2.1. Qualitative Analysis of Proteins

Table 2 shows the number of proteins identified by LC-MS/MS in individual samples and the number of proteins commonly found in all or all but one samples. We identified 235 proteins in untreated patients, 164 proteins in health controls, 301 proteins in the ERT-a group, and 222 proteins in the ERT-b group.

To analyze the distribution of proteins expressed in leukocytes across the different groups, a Venn diagram was generated (Figure 1). Appendix A shows this same list with the corresponding UniProt codes and names of the identified proteins.

These proteins were subjected to functional analysis using the FunRich program. In this analysis, we evaluated proteins implicated in the endosome, lysosomal lumen, lysosomal membrane, lysosome, mitochondria, and vesicles, since these organelles are affected in patients with MPS IVA [42,43,44] (Figure 2). The percentage of proteins involved in endosome function was lower in untreated versus control samples, and was higher in ERT-a and ERT-b samples than in untreated samples, although still lower than that of control samples. A similar pattern was observed in proteins expressed in the lysosomal membrane, mitochondrion, and vesicles. By contrast, the percentage of proteins expressed in the lysosome and lysosomal lumen was lower in ERT-a and ERT-b.

Further functional analyses performed using the Reactome program allowed us to identify the metabolic pathways in which these proteins participate. In samples from ERT treated patients we observed significant differences with respect to untreated patients in the expression of proteins involved in inflammation pathways, including of leukotrienes, eoxins, and thromboxane. Expression of proteins involved in the gluconeogenesis pathway was also partially normalized in MPS IVA patients that received ERT compared to untreated patients.

### 2.2. Quantitative Analysis of Proteins by SWATH

To identify dysregulated proteins in leukocyte samples, we performed a quantitative analysis using SWATH. To this end, we generated a protein library consisting of 690 potential biomarker proteins expressed by leukocytes in all groups analyzed. Next, we performed large-scale targeted protein analysis to quantify the levels of each protein in each group. In this analysis, we only considered dysregulated proteins with a *p*-value < 0.05 and a fold change (FC) > 1.4. Table 3 shows the number of dysregulated proteins found for each SWATH comparison. Also, see Appendix A.

#### 2.2.1. Proteins Down Regulated in MPS IVA Patients Relative to Healthy Controls

We next assessed the distribution of proteins that were downregulated in the untreated, ERT-a, and ERT-b groups relative to controls (Figure 3).

Table 4 shows the 91proteins that were down regulated in untreated patients relative to healthy controls, and indicates the changes observed in these proteins in the ERT-a and ERT-b groups. These proteins corresponded to the 36 proteins exclusively downregulated in the untreated group, the 15 downregulated proteins common to the untreated and ERT-a groups, the 6 proteins common to the untreated and ERT-b groups, and the 34 proteins common to the untreated, ERT-a, and ERT-b groups (Figure 3). We observed partial normalization (i.e., proteins remained dysregulated, *p* > 0.05) of the expression of proteins involved in the pyruvate metabolic pathway (KPYM, G3P, LDHA, PGK1, LDHB, ODPB and ALDOA; Table 4, gray rows) and complete normalization (protein expression was fully restored with treatment; *p* < 0.05) in some proteins involved in glucose metabolism (G6PI, PGM1 and G6PD; Table 4, green rows). Expression of most proteins involved in cytoskeletal organization (MOES, DEST, ARPC3, CORO1A, ANXA1, TPM1, K1C9, K22E, ARC1B, K2C1, K2C6A, TPM4, and K1C14) was normalized after ERT (Table 4, clear gray rows), although some remained downregulated (ARP2, ARPC5, ACTB). The following proteins are also involved in cytoskeletal organization: ADDB, a membrane-cytoskeleton-associated protein that promotes the assembly of spectrin-actin in a network of actin filaments (F-actin) and binds to actin monomers (G-actin); CNN2, which is involved in the regulation and modulation of smooth muscle contraction (Table 4, clear gray rows); SPTN1, which interacts with calmodulin in a calcium-dependent manner and regulates the movement of the cytoskeleton in the calcium-dependent membrane; VASP, which is involved in the remodeling of the cytoskeleton and all of them were normalized after ERT; and SEPT7, which is required for the normal organization of the actin cytoskeleton, and WDR1, which is involved in chemotactic cell migration by restricting the protrusions of the lamellipodial membrane, were further downregulated after ERT (Table 4, clear gray rows). Our findings reveal dysregulation of proteins involved in the lysosome/endosome system in untreated MPS IVA patients. Proteins involved in lysosomal membrane repair (LEG3 and VPS35) were normalized after ERT (Table 4, blue rows). Expression of STXB2 (Table 4, dark blue rows), which promotes vesicle trafficking and vesicle fusion with membranes in the SNARE position, was partially normalized in treated leukocytes (FC values decreased from 2.4 in untreated patients to 1.8 and 1. 6 in ERT-a or ERT-b groups, respectively). Expression of ubiquitin protein, UB2L3 (Table 4, clear blue rows), which is involved in vesicle trafficking from the endoplasmic reticulum to the lysosome, it was partially normalized. Expression of other proteins involved in membrane trafficking (ERP29 and PICAL; Table 4, dark blue rows) was also normalized after ERT.

We observed post-treatment normalization of the expression of several proteins involved in mitochondrial energy activity, assembly, or function (Table 4, pink rows) (ATPD, ATPA, CISY, PRDX5, ATPB), although others remained downregulated after ERT (ETFA, CH60, CH10, AATM, MDHM). TERA, which is involved in the formation of the transitional endoplasmic reticulum (tER), it was partially normalized only in the ERT-b group. GANAB, a catalytic subunit of glucosidase II of Glc2Man9GlcNAc2 oligosaccharide, acts as an immature precursor of glycoproteins involved in the *N*-glycan metabolism pathway, and was down regulated relative to healthy controls in all MPS IVA patient groups (Table 4, yellow rows). We also observed normalization after ERT of the expression of cytoplasmic and extracellular proteins (HBA, HBB) involved in oxygen transport and of proteins implicated in DNA binding (H13, H15). Proteins involved in the interconversion of metabolites, including hydrolases, dehydrogenases, transketolases, and isomerases (6PGD, ARGI1, TKT, THTR, PUR8, EM55, TTHY, IPYR2, DET1) were normalized after ERT (Table 4, dark gray rows). Among proteins involved in metabolic and cellular processes we observed normalization of PDIA3 and PSA; partial normalization of APT, PSME1, CD14, APOA2, PEBP1, and KAPCA; and no change in the expression of IGHG3 and EF1A3 (Table 4, dark pink rows). Several proteins with catalytic activity or transport-related functions showed partial normalization of expression after ERT (OLA1, RAN, CLIC1), although one such protein (VATA) showed no changes in expression (Table 4, orange rows). Proteins with DNA-binding functions (STIP1, PCBP2, HNRPK, ROA1, HMGB2) were partially or fully normalization after ERT (Table 4, clear red rows). Similarly, partial normalization of expression was observed for CEAM6, which belongs to the carcinoembryonic antigen-related cell adhesion molecule family, and OSTF1, which participates in bone resorption, interacts with secretion factors during bone formation, and participates in osteoclast development (Table 4, white rows).

Dysregulated proteins for which partial or complete normalization was observed after ERT are shown in the interaction diagrams in Figure 4. Figure 4A depicts the string network analysis of all 91 proteins that were downregulated in untreated MPS IVA patients relative to healthy controls. The results of the same analysis performed after treatment reveal a less complex network, given that many proteins were normalized by ERT.

Coronin 1A (CORO1A) is a crucial component of the cytoskeleton of highly mobile cells. This protein is involved in the invagination of large pieces of the plasma membrane and in the formation of protrusions of the plasma membrane, thereby contributing to cell locomotion. We carried out an interactome study using STING. As shown in Figure 5, we observed that CORO1A interacts with other specific proteins such as OSTF1, which participates in bone resorption and enhances osteoclast formation. In bone, CORO1A also regulates cathepsin K, which promotes degradation of collagen I and II.

Figure 6 shows the mean values of the area obtained for each individual sample in each group for CORO1A and GANAB proteins. GANAB is a catalytic subunit of glucosidase II that sequentially cleaves the two innermost alpha-1,3-linked glucose residues from the Glc(2)Man(9)GlcNAc(2) oligosaccharide precursor of immature glycoproteins. This protein is associated with the *N*-glycan metabolism pathway, which in turn is involved in glycan metabolism.

#### 2.2.2. Proteins Upregulated in Untreated MPS IVA Patients Relative to Untreated Patients and Controls

We next identified proteins that were upregulated in the healthy control, ERT-a, and ERT-b groups relative to the untreated MPS IVA group (Figure 7).

Table 5 lists the 73 proteins that were upregulated in MPS IVA patients relative to healthy controls. We quantified the FC in the expression of these proteins relative to healthy controls to determine whether their expression was normalized with treatment. Expression of proteins involved in the organization of cellular components (Table 5, light blue rows) was normalized after treatment (except for ARAP1). Similar effects were observed for proteins involved in the cellular response to stimulus (Table 5, light green rows) and proteins expressed in components of the lysosome (Table 5, light pink rows). Other proteins that showed normalization of expression after ERT included those involved in extracellular matrix (ECM) binding and cellular adhesion (Table 5, light orange rows), and in the function of organelles such as the endoplasmic reticulum, Golgi apparatus, and mitochondria (Table 5, purple rows). Similarly, ERT resulted in normalization of the expression of proteins expressed in secretory granules and in small subcellular vesicles formed in Golgi apparatus (Table 5, dark blue rows), and partial or complete normalization of proteins involved in transport (Table 5, dark green rows).

Normalization of the expression of proteins involved in the transit of vesicles to lysosomes was also observed (Table 5, purple rows), including RAB4A, which modulates the remodeling of the actin cytoskeleton; ARAP1, which mediates cholesterol biosynthesis and drug metabolism; NB5R3, which is normally found in lysosomes; and DEF3 and TRFL, which may be implicated in oxidative stress.

Figure 8 shows the mean values obtained in each group for TRLF (lactotransferrin) and VTNC (vitronectin) proteins, which are implicated in bone reabsorption and mineralization.

#### 2.2.3. Proteins Not Detected in Healthy Controls but Are Detected in MPSIV Patients

Table 6 shows proteins that were not detected in healthy controls and were downregulated in ERT-treated MPS IVA patients (ERT-a and/or ERT-b groups) relative to untreated patients. RNAS2 was common to the ERT-a and ERT-b groups. This protein is a non-secretory ribonuclease, which exerts selective chemotactic effects on dendritic cells. It is secreted by a range of innate immune cells, from blood cells to epithelial cells. Levels of this protein correlate with infection and inflammation processes. Recent studies have demonstrated that RNases in the extracellular space can exert immuno-modulatory effects.

## 3. Discussion

Variation in GALNS enzymatic activity has been proposed as a biomarker of MPS IVA: low levels are observed in more severe phenotypes and higher levels in attenuated forms [45,46]. However, enzymatic activity cannot be used as a biomarker to evaluate the therapeutic efficacy of ERT.

Other classical biomarkers in MPS IVA include levels of glycosaminoglycan (GAG). One such example is KS, which can be easily measured in blood and urine samples [3,45]. However, KS levels decrease with age and can be normalized in older patients with MPS IVA, especially after 12 years of age when the growth plate is entirely closed or destroyed [47,48]. C6S is another GAG that accumulates in MPS IVA, but is not widely used as a biomarker as it can be masked by other GAGs that appear in the same position in the MS/MS analysis spectrum [47]. The identification of definitive biomarkers of bone and cartilage disease in MPS IVA remains an unmet challenge, and is essential to monitor disease progression and treatment response.

Until now, no biomarker has been identified that can adequately reflect bone and cartilage pathology MPS IVA patients of all ages and at all disease stages. Detection of proteins that are dysregulated in MPS IVA constitutes a significant advance in our understanding of the mechanism underlying the disease. Using a proteomics-based approach, we have identified biomarkers with potential diagnostic and prognostic utility that are expressed in leukocytes in MPS IVA patients.

Previous reports have described the relationship between lysosomes with other organelles such as the transGolgi reticulum, mitochondrion, vesicles, and cytoskeleton [49,50,51,52,53]. Lysosomal activity is regulated by the activation of mTORC1 and CLEAR promotors [49]. The lysosome requires the cellular cytoskeleton in order to move and maintain its membrane structure. The main cellular impairment found in lysosomal disorders is the storage of undegraded substrates. This abnormal accumulation leads to alterations in proteins that participate in the interconnection between organelles. Ubiquitin proteins [54] and cholesterol [55] play important roles in this process; their principal function is the degradation or elimination of components and the biosynthesis of new ones [56]. Undegraded substrates can also modulate the function and location of cell receptors such as Toll-like receptors [57], activation of this receptor by ligands as AMP-activated protein kinase [58], It can reduce activity mTOR affection (this effect reduced the activation CLEAR promotors) [59] and insulin signaling [60], leading to cellular damage.

Using qualitative and quantitative proteomic methods to analyze proteins in leukocytes from MPS IV patients, we identified proteins that show altered expression relative to healthy controls, and then examined changes in their expression MPS IV patients that received ERT. Qualitative studies revealed that ERT resulted in normalization of dysregulated proteins in all parts of the cell. However, quantitative analyses showed only partial restoration of the expression of proteins involved in pyruvate metabolism, the cytoskeleton, vesicle trafficking, the mitochondrion, the Golgi-lysosome interaction, and repair of the lysosomal membrane, and no restorative effect on proteins involved in iron transport. Analysis of proteins involved in DNA binding showed that ERT resulted in partial normalization or had no effect. Interestingly, the inflammation pathway was activated only in samples from untreated MPS IV patients. In a previous study, we demonstrated qualitative and quantitative changes in proteins expressed by fibroblasts from MPS IVA patients following ERT and encapsulated ERT in nanoparticles [61].

The protein biomarker candidates identified in our quantitative analysis were TRFL, CORO1A, GANAB, and VTNC. In MPS IVA cellular inflammatory processes are affected. We demonstrated activation of inflammation factors in samples from untreated MPS IVA patients. Oxidative stress is caused by the accumulation of substrates in lysosomes. Therefore, the combination of anti-oxidants with ERT treatment can improve therapeutic efficacy [62]. We found that TRFL was upregulated in untreated MPS IVA patients compared with healthy controls, and that this effect was corrected after ERT. However, TRFL is easily dysregulated, possibly due to a lack of enzyme before the next dose of ERT is weekly administered. TRFL is transferred during lactation from mothers to infants, helping to improve the infant’s immune system, and exerts immune regulatory functions, decreasing the release of interleukin-1 (IL-1), IL-2, IL-6, IL-12, and tumor necrosis factor-α (TNF-α), and enhancing the cytotoxicity of natural killer cells and monocytes [63]. This protein is also implicated in iron metabolism [64] and bone regeneration [65]. In MPS IVA, the constant activation of this protein may be linked to more severe disease [66].

CORO1A is another protein that was downregulated in untreated MPS IVA patients compared with healthy controls, and was partially normalized after ERT. CORO1A regulates actin cytoskeleton-dependent processes (cytokinesis), cell polarization, migration, and phagocytosis, and plays an important role in Ca^2+^ signaling in macrophages [67]. These effects could correct the pro-inflammatory processes that occur in MPS IVA. In bone, CORO1A regulates cathepsin k [66,67,68,69] and exerts significant effects on bone resorption through degradation of bone-matrix proteins and type I and type II collagen [70]. The third protein identified that may serve as a biomarker in MPSIVA was GANAB. Although not well described, GANAB has one catalytic subunit of glucosidase II that sequentially cleaves 2 alpha-1,3-linked glucose residues from the Glc2Man9GlcNAc2 oligosaccharide, which in turn is a precursor of immature glycoproteins and participates in the *N*-glycan metabolic pathway [71]. VTNC is a glycoprotein predominantly produced by the liver, and expressed in the blood and ECM. VTNC, which binds to GAGs, collagen, plasminogen, and the urokinase-receptor, participates in stabilization of the inhibitory conformation of plasminogen activation inhibitor-1, and in the ECM can potentially regulate proteolytic degradation of this matrix. VTNC also participates in homeostatic processes, binding to complement, heparin, and thrombin–antithrombin III complexes, and exerts effects on the immune response. This protein can modulate multiple biological functions, influencing proteolytic enzyme activity and modulating protein kinases. In addition, the presence of the sequence RGD (Arg-Gly-Asp) in VTNC allows it to bind to the integrin receptor VnR (αvβ3) and modulate cell attachment [72]. αvβ3 is found in many cell types, including endothelial cells, chondrocytes, fibroblasts, monocytes, and activated blymphocytes [73]. In bone, VTNC is present in the bone matrix at low levels and at higher levels in the unmineralized osteoid, and participates in bone regeneration and mineralization. VTNC is degraded by the matrix metalloproteinases (MMP) collagenase-1 (MMP-1), gelatinase A (MMP-2), matrilysin (MMP-7), metalloelastase (MMP-12), and MT1-MMP (MMP-14) [74].

Given their involvement in bone metabolism, we consider TRFL, CORO1A, GANAB and VTNC to be candidate biomarkers of bone impairment in MPS IVA. We postulate that these proteins expressed in leukocytes may also be expressed in bone cells, given that macrophages and osteoclasts originate from the same cell line and may share similar proteins [65]. In a previous proteomic analysis of fibroblasts, we observed upregulation of TRFL in untreated MPS IVA patients and upregulation of GANAB in ERT-treated versus untreated MPS IVA patients [61].

One protein that was downregulated in MPS IVA patients that received ERT was RNAS2. This specific pyrimidine nuclease shows a slight preference for cytotoxin and helminthotoxin, is located in the lysosome, and is described as a chemotactic factor for dendritic cells. Several other proteins showed partial or complete normalization of expression after ERT, including proteins implicated in the pyruvate metabolic pathway (LDHA, LDHB) and in cytoskeletal organization (MOES) [61]. Inflammation-associated proteins that were dysregulated in untreated patients included OLFM4, TGFB1, THAS, PGH1, CAP7, andCHIT1 [70,75]. PRTN3, which is involved in collagen I and II degradation, was also dysregulated in untreated patients, in line with the well described impairment of collagen I and II biosynthesis and catabolism in MPS IVA [76,77].

A qualitative study conducted using the Funrich program produced similar results to our quantitative analysis in terms of ERT-mediated normalization of dysregulated protein expression. While the quantitative study allowed precise quantification of proteins whose expression was normalized after ERT, the qualitative study gives us only an identification of proteins expressed in all conditions combining these two approaches allowed us to confirm the effect of ERT on the expression of dysregulated proteins in MPS IVA [76,77].

### 3.1. Limitations of the Study

In our cohort we identified and analyzed 690 proteins with 99% reliability. The SWATH method is a specific, reproducible, and sensitive approach, allowing relative or absolute protein quantification. However, some limitations of this approach should be noted. First, the number of proteins identified/quantified is largely limited by the composition of the spectral library. In this study, the library was generated with a pool of proteins for each condition, and consisted of a total of 690 proteins, all with a FDR < 1%. In contrast to the selected reaction monitoring (SRM) technique, in which only three transitions are quantified, in the SWATH technique, seven transitions are quantified for each peptide. Therefore, 10 peptides per protein must be identified in order to extract the peak areas necessary for quantification. Despite the precision of this technique, this also constitutes a limitation, as proteins in the library for which less than 10 peptides are identified will not be quantified. Due to this limitation, the SWATH method may be unable to detect collagens, pro-inflammatory factors and lysosomal enzymes that degrade substrates in the lysosome. Interestingly, our analysis detected TRFL, which modulates interleukins, and PRTN3, which degrades collagen, but failed to detect any collagens, interleukins, or lysosomal enzymes.

Another limitation of our study is that normalization of protein expression in leukocytes does not necessarily correlate with normalization in chondrocytes in the avascular region. The enzyme is easily taken up by the leukocytes, restoring the function of dysregulated metabolic pathways in these cells while the infused enzyme circulates in the blood. It is critical to understand whether these proteins are dysregulated in bone and cartilage in MPS IVA patients in order to identify potential diagnostic biomarkers of disease severity. When a bone-penetrating drug becomes available, surrogate biomarkers will be essential.

### 3.2. Conclusions

Lysosomal disorders are characterized by significant alterations in proteins caused by the accumulation of undegraded substrate. We have identified a large set of proteins that are dysregulated in leukocytes from untreated MPS IVA patients, and many of which are fully or partially normalized following ERT. Several of these proteins are implicated in bone metabolism and are therefore potential biomarkers of the severity of bone disease in MPS IVA. These candidate proteins should be investigated in bone and cartilage specimens in MPS IVA patients to determine the extent to which they truly reflect bone pathology.

## 4. Materials and Methods

### 4.1. Study Work Flow

The graphic below outlines the workflow used to identify dysregulated proteins in leukocytes. Analyses consisted of both qualitative (LC-MS/MS) and quantitative (SWATH-MS) proteomic methods (Figure 9).

Blood samples were collected from 3 groups of participants: untreated MPS IVA patients; ERT-treated MPS IVA patients (sampled before ERT and 1day after), and healthy controls. Blood samples were separated into plasma and leukocytes. The leukocytes were lysed, and proteomic analyses performed. The workflow in Figure 9 depicts the qualitative analysis (left), which identified proteins expressed in leukocytes, and the quantitative analysis (right), which determined the number of specific proteins expressed in leukocytes. After these two distinct proteomics approaches, bioinformatics analyses were performed to obtain more information about the identified/quantified proteins.

### 4.2. Samples

Cell samples for proteomic analyses were obtained from MPS IVA patients in 3 hospitals in Spain after receiving informed consent: 8 untreated patients and 5 patients who underwent weekly ERT. In the ERT group, blood samples were obtained before (ERT-a) and 24 h after (ERT-b) treatment. All patients presented the classical MPS IVA phenotype. For the control group, blood samples were obtained from 6 healthy donors.

### 4.3. Protein Extraction

Leukocytes were sonicated to rupture the membrane and release proteins, and then centrifuged for 10 min at 10,000× rpm and 4 °C. Protein extracts were recovered from the supernatant and subsequently frozen at −20 °C.

### 4.4. Enzyme Activity Test

An enzyme activity test was used to analyze GALNS enzymatic activity in samples [78]. Results are shown in Table 7 (physiologicalrange, 1.8–20.0 nM/h/mg).

### 4.5. Proteomic Analysis

Protein identification and quantification were performed as described in bibliography [61,79,80,81,82]. For protein identification, digested peptides from each sample were separated using reverse phase chromatography. The gradient was developed using a micro liquid chromatography system (Eksigent Technologies nanoLC 400, Sciex, Redwood City, CA, USA) coupled to a high-speed Triple TOF 6600 mass spectrometer (Sciex, Redwood City, CA, USA) with a microflow source. The analytical column used was a Chrom XP C18 silica-based reversed-phase column (150 × 0.30 mm) with a 3mm particle size and 120Å pore size (Eksigent, Sciex Redwood City, CA, USA). The trap column was a YMC-TRIART C18 (YMC Technologies Teknokroma Analítica, Barcelona, Spain), with a 3mm particle size and 120Å pore size, that was switched on-line with the analytical column. Data were acquired with a TripleTOF 6600 System (Sciex, Redwood City, CA, USA) using a data-dependent workflow (DDA).

For the SWATH analysis a spectral library was created using pooled samples from each group (healthy controls, untreated patients, and the ERT-a and ERT-b groups) using a DDA. Next, peak extraction was performed using the MS/MSALL add-in for PeakView Software (v. 2.2., Sciex, Redwood City, CA, USA) with the SWATH Acquisition MicroApp (v. 2.0., Sciex, Redwood City, CA, USA). Only peptides with a confidence score > 99% (as obtained from a Protein Pilot database search) were included in the spectral library.

SWATH–MS acquisition was performed on a TripleTOF^®^ 6600 LC-MS/MS system (Sciex, Redwood City, CA, USA). Samples from each group were analyzed using the data-independent acquisition (DIA) method. Targeted data extraction of the fragment ion chromatogram traces from the SWATH runs was performed in PeakView (v. 2.2) using the SWATH Acquisition MicroApp (v. 2.0). The integrated peak areas (processed mrkvw files from PeakView) were directly exported to MarkerView software (Sciex, Redwood City, CA, USA) for relative quantitative analysis.

Unsupervised multivariate statistical analysis using principal component analysis (PCA) was performed to compare data across the samples using a range scale. The average MS peak area for each protein was derived from the biological replicates of the SWATH-MS of each sample, followed by analysis using a Student’s *t*-test (MarkerView software, sciex, Redwood City, CA, USA) to compare between samples based on the averaged total area of all transitions for each protein. The *t*-test result (*p*-value) indicates how well each variable distinguishes the two groups. Candidate proteins were selected for each library based on *t*-test results (*p*-value < 0.05 and FC (increase or decrease) > 1.4).

Functional analysis was performed by FunRich (Functional Enrichment analysis tool) open-access software for functional enrichment and interaction network analysis (http://funrich.org/index.html).Reactome (https://reactome.org). STRING (https://string-db.org).PANTHER (http://pantherdb.org).

## Figures and Tables

**Figure 1 ijms-22-00226-f001:**
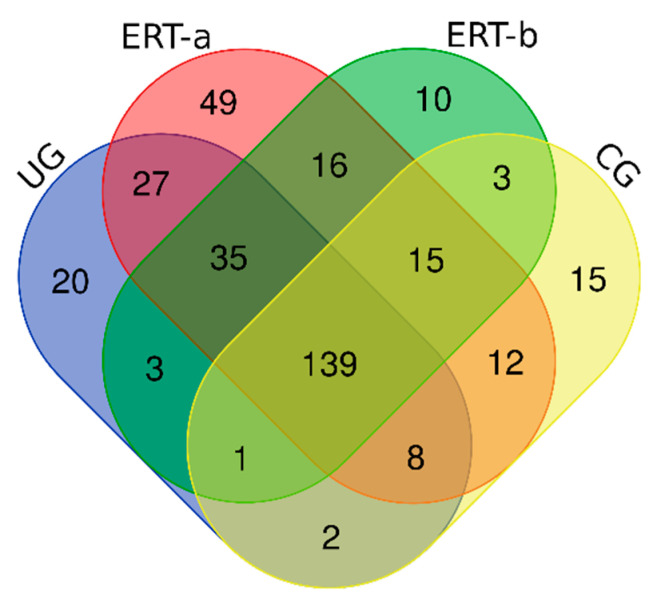
Venn diagram showing the distribution of proteins in the 4 study groups. Abbreviations: CG: control group; ERT-a, patients sampled before enzyme infusion; ERT-b, patients sampled 24 h after enzyme infusion; UG: untreated group.

**Figure 2 ijms-22-00226-f002:**
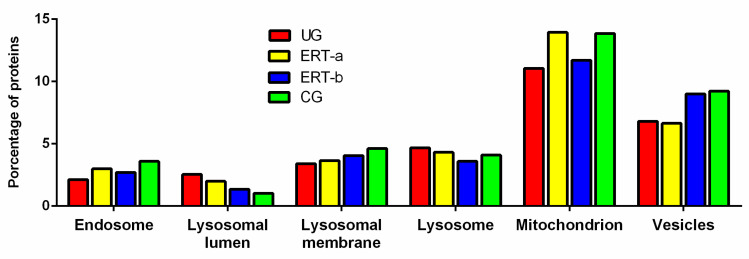
Functional analysis using the FunRich program. This figure shows the changes in the relative percentages of leukocyte proteins expressed in the endosome, different parts of the lysosome, the mitochondria, and the vesicles in each of the 4 groups. Abbreviations: CG: control group; ERT-a, patients sampled before enzyme infusion; ERT-b, patients sampled 24 h after enzyme infusion; UG: untreated group.

**Figure 3 ijms-22-00226-f003:**
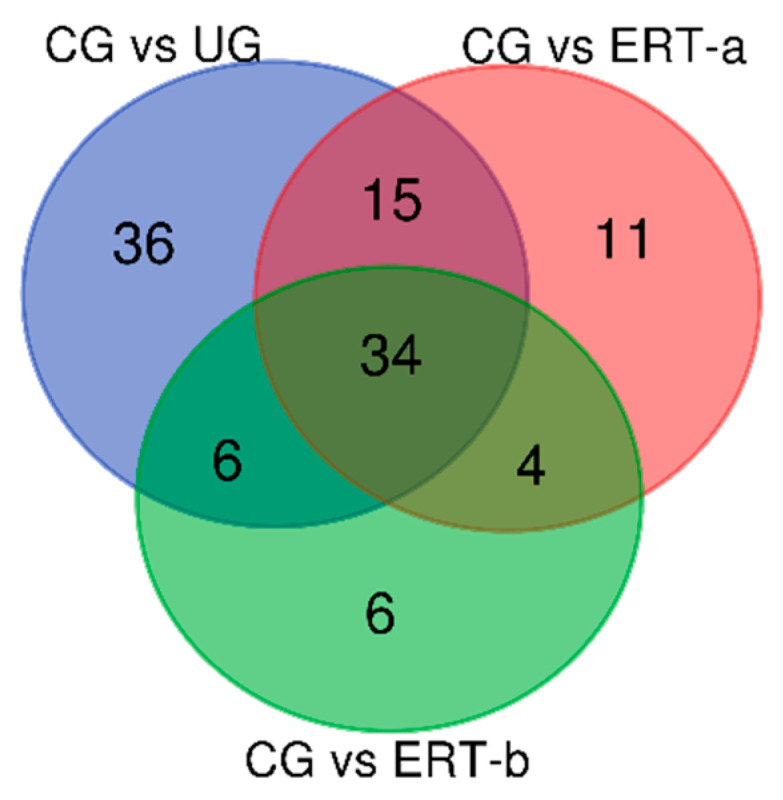
Venn diagram showing the distribution of leukocyte proteins that were downregulated the untreated, ERT-a, and ERT-b groups relative to healthy controls. Abbreviations: CG, control group; UG, untreated group; ERT-a, patients sampled before enzyme infusion; ERT-b, patients sampled 24 h after enzyme infusion.

**Figure 4 ijms-22-00226-f004:**
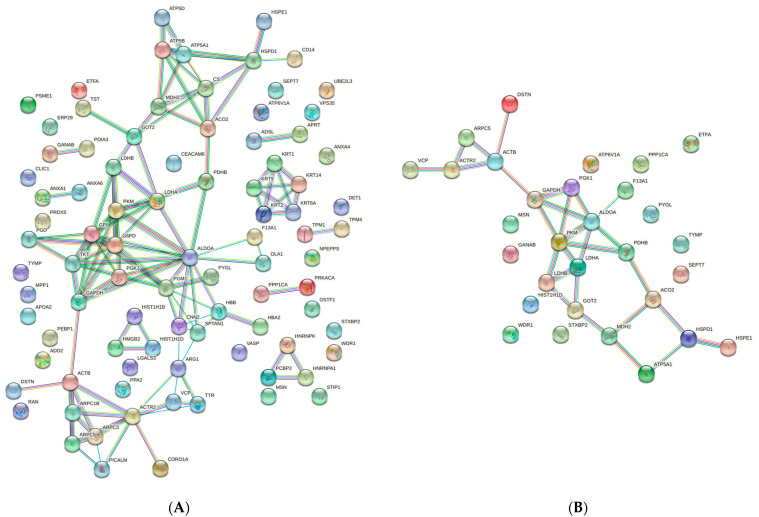
String network analysis. (**A**) Proteins downregulated in untreated MPS IVA patients. (**B**) Proteins (*n* = 32) that remain dysregulated or partially dysregulated in ERT-treated MPS IVA patients. Abbreviations: MPS IVA, Mucopolysaccharidosis type IVA; ERT, enzyme replacement therapy.

**Figure 5 ijms-22-00226-f005:**
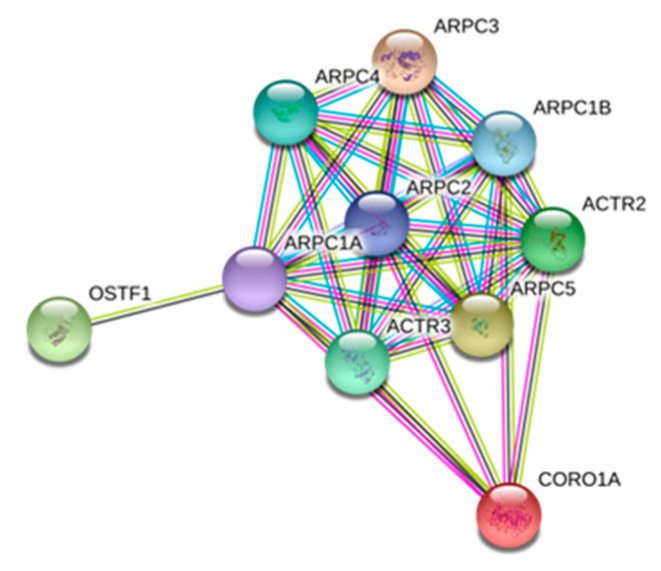
STRING analysis of the CORO1A interactome. Interactions among proteins found CORO1A and OSTF1, where participated ARP family proteins.

**Figure 6 ijms-22-00226-f006:**
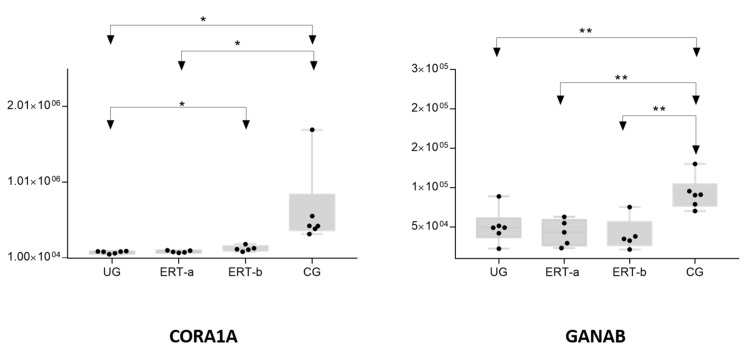
Changes in the levels of CORO1A and GANAB proteins in the different groups analyzed. Each point represents the mean value of the protein area for an individual sample in each group. The bottom and top of each box represent the first and third quartiles, respectively. The whiskers represent the minimum and maximum values within 1.5 times the interquartile range. Any data points not included between the whiskers are considered outliers. * *p* < 0.05; ** *p* < 0.01. Abbreviations: CG, control group; ERT-a: ERT-a, patients sampled before enzyme infusion; ERT-b, patients sampled 24 h after enzyme infusion; UG: untreated group.

**Figure 7 ijms-22-00226-f007:**
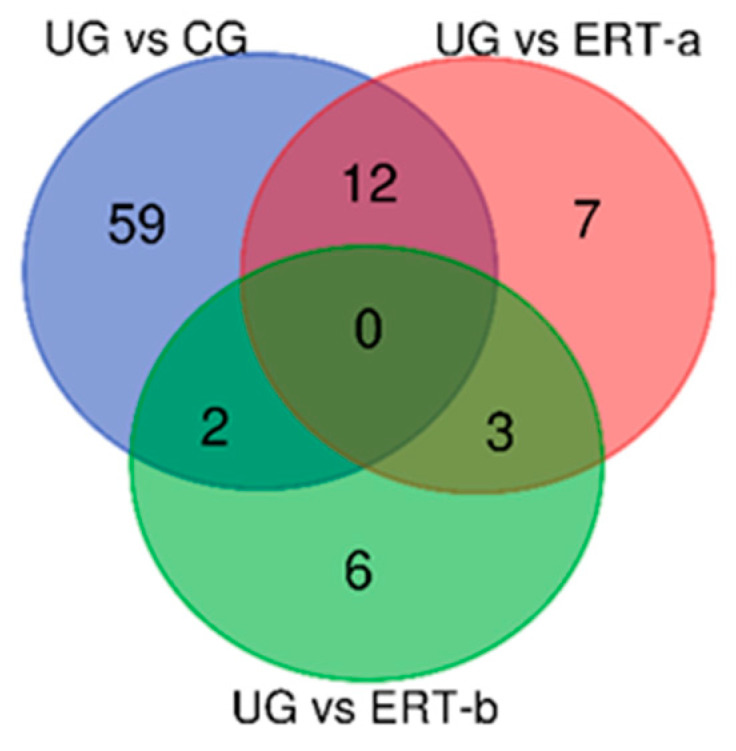
Venn diagram showing the distribution of leukocyte proteins that were the healthy control. Abbreviations: CG, control group; UG, untreated group; ERT-a, patients sampled before enzyme infusion; ERT-b, patients sampled 24 h after enzyme infusion.

**Figure 8 ijms-22-00226-f008:**
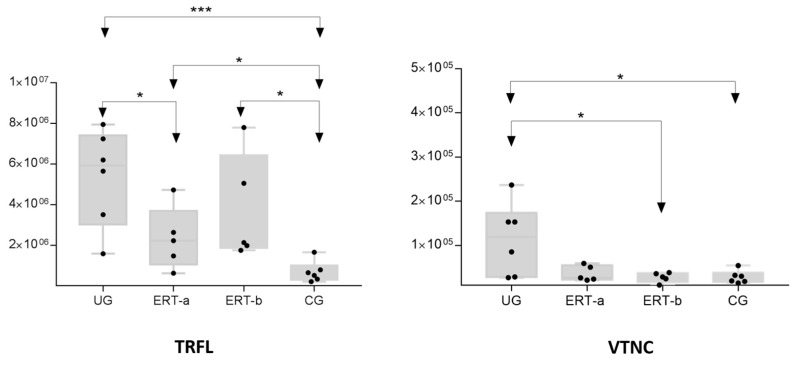
Variations in the levels of TRFL and VTNC proteins in the different groups analyzed. Each point represents the mean value of the protein area for an individual sample in each group. The bottom and top of each box represent the first and third quartiles, respectively. The whiskers represent the minimum and maximum values within 1.5 times the interquartile range. Any data points not included between the whiskers are considered outliers. * *p* < 0.05; *** *p* < 0.001. Abbreviations: CG, control group; ERT-a: ERT-a, patients sampled before enzyme infusion; ERT-b, patients sampled 24 h after enzyme infusion; TRFL, lactotransferrin; UG: untreated group; VTNC, vitronectin.

**Figure 9 ijms-22-00226-f009:**
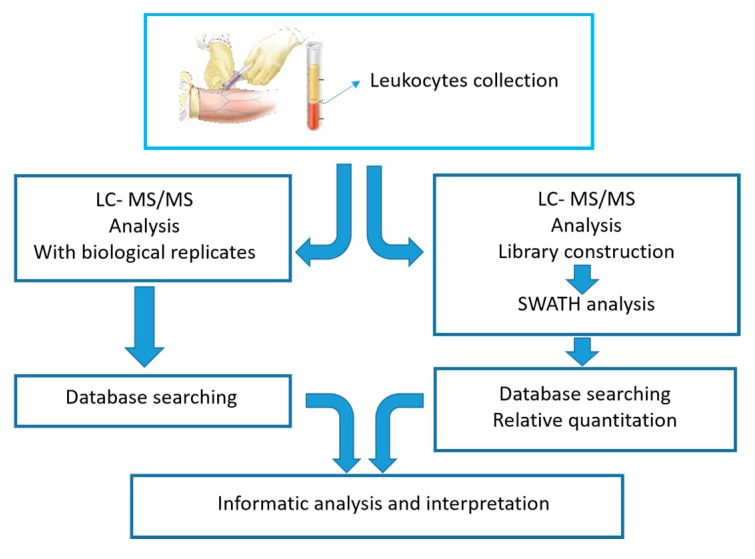
Schematic illustrating the study workflow showing both qualitative (LC-MS/MS, left) and quantitative (SWATH-MS, right) proteomic methods.

**Table 1 ijms-22-00226-t001:** Demographics of MPS IVA patients.

Patient ID	Sex	Age at Diagnosis (y)	ERT	Age at Start of Treatment (y)	Current Characteristics				
					Age (y)	Height (cm)	6 Minute Walk Test (m)	FVC (mL)	FEV_1_ (mL)
1	F	1	No	-	31	98	250	600	500
2	M	2	No	-	31	113	305	870	700
3	M	2	No	-	21	95	ND *	380	260
4	M	2	No		40	99	ND *	480	360
5	F	4	No	-	15	103	341	920	820
6	F	3	No	-	29	99	ND *	ND **	ND **
7	F	1	No	-	18	119	272	110	900
8	M	unknown	No	-	21	103	ND *	920	700
9	M	1	Yes	12	16	100	105	690	450
10	M	2	Yes	2	6	104	450	770	720
11	M	3	Yes	13	18	113.5	472	1390	1330
12	M	3	Yes	11	19	113	234	1350	1160
13	M	5	Yes	18	22	110	344	870	730

Patients 1 and 2 are twin brothers; patients 6 and 7 are brothers * patient cannot walk; ** patient undergoing ventilation with tracheostomy. Abbreviations: ERT, enzyme replacement therapy; F, female; FEV_1_, forced expiratory volume in 1 s; FVC, forced vital capacity; ID, identification, M, male; ND, not determined.

**Table 2 ijms-22-00226-t002:** Number of proteins per sample and number of common proteins per group. Only proteins with FDR < 1% were selected.

Patients and Control Groups-	Sample ID	Proteins Identified Per Sample (*n*)	Proteins Identified in All or All but One Samples (*n*)
Untreated Group	UG 1	460	235
	UG 2	330	
	UG 3	367	
	UG 4	612	
	UG 5	NA	
	UG 6	NA	
	UG 7	341	
	UG 8	304	
ERT-a Group	ERT-a 1	480	301
	ERT-a 2	338	
	ERT-a 3	369	
	ERT-a 4	406	
	ERT-a 5	400	
ERT-b Group	ERT-b 1	492	222
	ERT-b 2	190	
	ERT-b 3	437	
	ERT-b 4	252	
	ERT-b 5	557	
Healthy Control Group	CG 1	144	164
	CG 2	470	
	CG 3	238	
	CG 4	286	
	CG 5	315	
	CG6	261	

Abbreviations: ERT-a, patients sampled before enzyme infusion; ERT-b, patients sampled 24 h after enzyme infusion; NA, not analyzed (due to very low quantity of leukocytes).

**Table 3 ijms-22-00226-t003:** Dysregulated proteins in leukocyte samples from MPSIVA patients. Proteins considered dysregulated are those with a *p*-value < 0.05 and a fold change (FC) > 1.4.

Comparison	Proteins Downregulated Compared with Controls	Proteins Upregulated Compared with Controls
Control vs. Untreated	91	73
Control vs. ERT-a	64	55
Control vs. ERT-b	49	56
Comparison	Proteins down regulated compared with untreated group	Proteins upregulated compared with untreated group
Untreated vs. ERT-a	22	10
Untreated vs. ERT-b	10	23
Comparison	Proteins down regulated compared with ERT-a group	Proteins upregulated compared with ERT-b group
ERT-a vs. ERT-b	4	12

Abbreviations: ERT-a, patients sampled before enzyme infusion; ERT-b, patients sampled 24 h after enzyme infusion.

**Table 4 ijms-22-00226-t004:** Downregulated proteins in untreated and ERT-treated MPS IVA patients (ERT-a, ERT-b) with respect to healthy controls. Proteins considered dysregulated are those with a *p*-value < 0.05 and a fold change (FC) > 1.4.

UniProt Code	UniProt Name	Protein Name	Fold Change Relative to Healthy Controls		
			Untreated	ERT-a	ERT-b
P14618	KPYM	Pyruvate kinase PKM	0.0537	0.0770	0.1210
P04406	G3P	Glyceraldehyde-3-phosphate dehydrogenase	0.0539	0.0729	0.0632
P00338	LDHA	l-lactate dehydrogenase A chain	0.0629	0.1039	0.1176
P00558	PGK1	Phosphoglycerate kinase 1	0.0902	0.1124	0.1742
P07195	LDHB	l-lactate dehydrogenase B chain	0.2261	0.2049	0.2358
P11177	ODPB	Pyruvate dehydrogenase E1 component subunit beta. mitochondrial	0.3005	0.2520	0.3246
P04075	ALDOA	Fructose-bisphosphatealdolase A	0.4012	0.5957	0.6128
P06744	G6PI	Glucose-6-phosphate isomerase	0.1645	0.3401	0.3358
P36871	PGM1	Phosphoglucomutase-1	0.3284	0.2524	0.5816
P11413	G6PD	Glucose-6-phosphate 1-dehydrogenase	0.3951	0.4585	0.5714
P26038	MOES	Moesin	0.1282	0.1284	0.2848
P60981	DEST	Destrin	0.3604	0.2756	0.4310
O15145	ARPC3	Actin-related protein 2/3 complex subunit 3	0.5094	0.4480	0.5276
P31146	CORO1A	Coronin-1A	0.1283	0.1418	0.2020
P09493	TPM1	Tropomyosin alpha-1 chain	0.2997	0.6517	0.7882
P35527	K1C9	Keratin. type I cytoskeletal 9	0.3019	0.5949	0.6857
P35908	K22E	Keratin. type II cytoskeletal 2 epidermal	0.2940	0.6149	0.8652
O15143	ARC1B	Actin-related protein 2/3 complex subunit 1B	0.3132	0.4472	0.4072
P04264	K2C1	Keratin. type II cytoskeletal 1	0.3301	0.6055	0.7358
P02538	K2C6A	Keratin. type II cytoskeletal 6A	0.4352	1.0918	2.2909
P67936	TPM4	Tropomyosin alpha-4 chain	0.4602	0.8777	0.5522
P02533	K1C14	Keratin. type I cytoskeletal 14	0.5186	0.8316	0.7752
P61160	ARP2	Actin-related protein 2	0.3608	0.3315	0.3566
O15511	ARPC5	Actin-related protein 2/3 complex subunit 5	0.4050	0.5206	0.4899
P60709	ACTB	Actin. cytoplasmic 1	0.3010	0.2164	0.2500
Q99439	CNN2	Calponin-2	0.5653	1.2121	0.9130
P35612	ADDB	Beta-adducin	0.4505	0.5356	0.7874
Q13813	SPTN1	Spectrin alpha chain. non-erythrocytic 1	0.6223	0.9501	0.9308
P50552	VASP	Vasodilator-stimulated phosphoprotein	0.4809	0.7258	0.6589
O75083	WDR1	WD repeat-containing protein 1	0.6081	0.5059	0.4914
Q16181	SEPT7	Septin-7	0.3998	0.2507	0.3119
P17931	LEG3	Galectin-3	0.3903	0.4969	0.6100
Q96QK1	VPS35	Vacuolar protein sorting-associated protein 35	0.4053	0.3453	0.7819
Q15833	STXB2	Syntaxin-binding protein 2	0.4043	0.5382	0.6145
P68036	UB2L3	Ubiquitin-conjugating enzyme E2 L3	0.3696	0.6336	0.5254
P30040	ERP29	Endoplasmic reticulum resident protein 29	0.5751	0.8206	0.6180
Q13492	PICAL	Phosphatidylinositol-binding clathrin assembly protein	0.6298	0.6427	0.6980
P30049	ATPD	ATP synthasesubunit delta. mitochondrial	0.5906	0.8645	0.9240
P25705	ATPA	ATP synthasesubunit alpha. mitochondrial	0.3060	0.2776	0.5251
O75390	CISY	Citrate synthase. mitochondrial	0.6794	0.5843	0.6636
P30044	PRDX5	Peroxiredoxin-5. mitochondrial	0.4486	0.6043	0.5455
P06576	ATPB	ATP synthase subunit beta. mitochondrial	0.4890	0.7480	0.7076
Q99798	ACON	Aconitatehydratase. mitochondrial	0.5100	0.4129	0.6735
P13804	ETFA	Electron transfer flavoprotein subunit alpha. mitochondrial	0.2472	0.1565	0.1542
P10809	CH60	60 kDa heat shock protein. mitochondrial	0.1905	0.4708	0.3593
P61604	CH10	10 kDa heat shock protein. mitochondrial	0.3779	0.3537	0.3176
P00505	AATM	Aspartate aminotransferase. mitochondrial	0.5693	0.4255	0.3930
P40926	MDHM	Malate dehydrogenase. mitochondrial	0.6669	0.5397	0.6540
P69905	HBA	Hemoglobin subunit alpha	0.2772	0.6311	0.4201
P68871	HBB	Hemoglobin subunit beta	0.2063	0.6678	0.3896
P55072	TERA	Transitional endoplasmic reticulum ATPase	0.2347	0.1977	0.3911
Q14697	GANAB	Neutral alpha-glucosidase AB	0.5454	0.4364	0.4613
P08133	ANXA6	Annexin A6	0.4899	0.7086	0.9723
P09525	ANXA4	Annexin A4	0.4646	0.6825	0.7587
P04083	ANXA1	Annexin A1	0.2010	0.5650	0.6752
P52209	6PGD	6-phosphogluconate dehydrogenase decarboxylating	0.1136	0.1570	0.2218
P05089	ARGI1	Arginase-1	0.2834	0.4069	0.4421
P29401	TKT	Transketolase	0.3432	0.3358	0.2993
Q16762	THTR	Thiosulfate sulfurtransferase	0.4924	0.7856	0.8081
P30566	PUR8	Adenylosuccinatelyase	0.2364	0.1361	0.7371
Q00013	EM55	55 kDa erythrocyte membrane protein	0.4414	0.4275	0.6262
P02766	TTHY	Transthyretin	0.4808	0.4167	0.5573
Q9H2U2	IPYR2	Inorganic pyrophosphatase2.mitochondrial	0.5533	0.6667	0.7373
Q7L5Y6	DET1	DET1 homolog	0.3518	0.5613	0.6315
P19971	TYPH	Thymidine phosphorylase	0.2521	0.2706	0.4983
P00488	F13A	Coagulation factor XIII A chain	0.2992	0.3087	0.3043
P06737	PYGL	Glycogen phosphorylase. liver form	0.1167	0.1441	0.2094
P62136	PP1A	Serine/threonine-protein phosphatase PP1-alpha catalytic subunit	0.5690	0.4578	0.5951
P30101	PDIA3	Protein disulfide-isomerase A3	0.5598	1.0088	1.1788
P55786	PSA	Puromycin-sensitive aminopeptidase	0.4389	0.6256	0.6560
P07741	APT	Adenine phosphoribosyltransferase	0.2659	0.2516	0.3201
Q06323	PSME1	Proteasome activator complexsubunit 1	0.3035	0.4744	0.4342
P08571	CD14	Monocyte differentiation antigen CD14	0.4434	0.3594	1.0551
P02652	APOA2	Apolipoprotein A-II	0.3696	0.2974	0.8084
P30086	PEBP1	Phosphatidylethanolamine-binding protein 1	0.5324	0.6353	0.8049
P17612	KAPCA	cAMP-dependent protein kinase catalytic subunit alpha	0.2921	0.2831	0.4253
P01860	IGHG3	Immunoglobulin heavy constant gamma 3	0.5132	0.4999	0.4892
Q5VTE0	EF1A3	Putative elongation factor 1-alpha-like 3	0.3466	0.3686	0.4249
Q9NTK5	OLA1	Obg-like ATPase 1	0.1099	0.3431	0.2186
P62826	RAN	GTP-binding nuclear protein Ran	0.1277	0.1178	0.3363
O00299	CLIC1	Chloride intracellular channel protein 1	0.2541	0.4800	0.6103
P38606	VATA	V-type proton ATPase catalytic subunit A	0.3386	0.4063	0.3411
P31948	STIP1	Stress-induced-phosphoprotein 1	0.3071	0.5992	0.5741
Q15366	PCBP2	Poly(rC)-binding protein 2	0.4584	0.3282	0.4541
P61978	HNRPK	Heterogeneous nuclear ribonucleoprotein K	0.5087	0.5807	0.6304
P09651	ROA1	Heterogeneous nuclear ribonucleoprotein A1	0.5798	0.6174	0.6532
P26583	HMGB2	High mobility group protein B2	0.1873	0.4159	0.6264
P16402	H13	Histone H1.3	0.0761	0.2883	0.2277
P16401	H15	Histone H1.5	0.0896	0.7384	1.0462
P40199	CEAM6	Carcinoembryonic antigen-related cell adhesion molecule 6	0.2537	0.4984	0.7378
Q92882	OSTF1	Osteoclast-stimulating factor 1	0.2963	0.6296	0.9094

Fold-change values in red indicate proteins that were not significantly dysregulated compared with healthy controls (*p* > 0.05). For more details, Appendix A. Protein related to metabolic pathway of pyruvate is shown in grey, glucose metabolism in green, cytoskeleton organization in clear gray, lysosome membranes reparation in blue, vesicle traffic and vesicle fusion in dark blue, oxygen transport in dark pink, mitochondrial organelle in pink, *N*-glycan metabolism pathway in yellow, DNA binding in clear orange, metabolite interconversion activity in dark green, metabolic and cellular process in dark orange, catalytic activity or proteins related to transport in purple and other proteins in white.

**Table 5 ijms-22-00226-t005:** Proteins upregulated in untreated and ERT-treated MPS IVA patients relative to healthy controls. Proteins considered dysregulated are those with a *p*-value < 0.05 and a fold change (FC) > 1.4.

UniProt Code	UniProt Name	Protein Name	Fold change Relative to Healthy Controls		
			Untreated	ERT-a	ERT-b
P30405	PPIF	Peptidyl-prolylcis-transisomerase F. mitochondrial	1.8656	0.9764	1.2728
P62318	SMD3	Small nuclear ribonucleoproteinSm D3	2.1112	1.8227	1.5860
Q9H4B7	TBB1	Tubulin beta-1 chain	2.6765	1.4412	1.5400
P61224	RAP1B	Ras-related protein Rap-1b	2.8033	1.2182	1.1940
P14780	MMP9	Matrix metalloproteinase-9	3.1051	1.7732	1.5370
P08567	PLEK	Pleckstrin	1.4669	1.0482	1.0934
Q6DRA6	H2B2D	Putative histone H2B type 2-D	8.3208	2.3169	1.3557
Q9BTM1	H2AJ	Histone H2A.J	8.8293	2.3197	1.3487
Q99879	H2B1M	Histone H2B type 1-M	9.9891	2.1137	1.4663
P62805	H4	Histone H4	12.0044	3.0822	2.1018
Q6UX71	PXDC2	Plexin domain-containing protein 2	2.0905	1.0293	1.0988
Q8WWA1	TMM40	Transmembrane protein 40	4.9507	1.5721	1.2002
P62314	SMD1	Small nuclear ribonucleo protein Sm D1	2.3589	1.9495	1.7400
P20338	RAB4A	Ras -related protein Rab-4A	1.5322	2.1434	1.0633
Q96P48	ARAP1	Arf-GAP with Rho-GAP domain. ANK repeat and PH domain-containing protein 1	2.036	1.8559	1.3848
P01137	TGFB1	Transforming growth factor beta-1 proprotein	2.336	1.0245	1.2298
P05106	ITB3	Integrin beta-3	1.6777	1.0676	0.9028
P02775	CXCL7	Platelet basic protein	3.1669	1.0947	1.0450
P41218	MNDA	Myeloid cell nuclear differentiation antigen	3.2656	1.7116	0.7412
P02776	PLF4	Platelet factor 4	3.9596	1.3654	1.3027
P11234	RALB	Ras-related protein Ral-B	2.004	1.1184	1.0789
P12838	DEF4	Neutrophil defensin 4	2.4608	2.3711	1.2386
P59666	DEF3	Neutrophil defensin 3	8.9484	3.7253	1.5188
O14773	TPP1	Tripeptidyl-peptidase 1	1.4441	1.1461	1.0245
Q8NBS9	TXND5	Thioredoxin domain-containing protein 5	1.5722	1.0267	1.2345
P24158	PRTN3	Myeloblastin	2.0115	1.8480	1.4986
P50990	TCPQ	T-complexprotein 1 subunittheta	2.0416	1.2148	1.2408
P12724	ECP	Eosinophil cationic protein	2.4191	1.0259	0.5594
P17213	BPI	Bactericidal permeability-increasing protein	3.3966	1.5635	1.2036
P05164	PERM	Myeloperoxidase	5.511	2.0122	1.2871
P20160	CAP7	Azurocidin	7.2103	2.0481	1.1568
Q13231	CHIT1	Chitotriosidase-1	2.1411	2.0308	1.7411
P00387	NB5R3	NADH-cytochrome b5 reductase 3	2.4463	1.7933	1.2666
P23229	ITA6	Integrin alpha-6	1.8854	1.0913	0.6926
P21926	CD9	CD9 antigen	1.9864	1.0081	0.9969
Q9Y6C2	EMIL1	EMILIN-1	2.4765	1.1540	1.2063
P04004	VTNC	Vitronectin	4.0358	3.1389	4.1505
P07996	TSP1	Thrombospondin-1	2.0331	1.0455	1.0711
Q6UX06	OLFM4	Olfactomedin-4	2.8132	2.4717	2.1722
Q15084	PDIA6	Protein disulfide-isomerase A6	1.4635	1.0387	1.1286
P04839	CY24B	Cytochrome b-245 heavy chain	1.5946	1.6427	1.2274
Q9HDC9	APMAP	Adipocyte plasma membrane-associated protein	1.615	1.3920	1.3258
P61769	B2MG	Beta-2-microglobulin	1.6224	1.1067	0.8221
P24557	THAS	Thromboxane-A synthase	1.7477	1.6320	1.4601
P04844	RPN2	Dolichyl-diphospho-oligosaccharide—proteinglycosyltransferasesubunit 2	1.8642	1.3634	1.2661
Q9NQC3	RTN4	Reticulon-4	1.9867	1.5387	1.0570
Q14165	MLEC	Malectin	2.1586	1.0898	0.9708
Q9BSJ8	ESYT1	Extended synaptotagmin-1	2.4649	1.2970	1.2265
Q8TC12	RDH11	Retinol dehydrogenase 11	3.061	1.3400	0.8831
P23219	PGH1	Prostaglandin G/H synthase 1	3.6929	1.5669	1.3489
P02774	VTDB	Vitamin D-binding protein	2.4327	1.4855	1.6653
P41240	CSK	Tyrosine-protein kinase CSK	3.4288	1.8739	3.5935
P02749	APOH	Beta-2-glycoprotein 1	5.374	2.4436	3.0788
Q00325	MPCP	Phosphate carrier protein. mitochondrial	2.2758	2.5252	1.3470
Q9UFN0	NPS3A	ProteinNip Snap homolog 3ª	1.6409	1.3436	1.3143
Q96P48	ARAP1	Arf-GAP with Rho-GAP domain. ANK repeat and PH domain-containing protein 1	2.036	1.8559	1.3848
P20061	TCO1	Transcobalamin-1	1.7894	1.2442	1.2795
P80188	NGAL	Neutrophil gelatinase-associated lipocalin	1.983	1.5798	1.2060
P08246	ELNE	Neutrophil elastase	2.9474	2.0215	0.9159
P00747	PLMN	Plasminogen	3.3382	1.2731	1.4806
Q8NBM8	PCYXL	Prenylcysteine oxidase-like	3.8514	1.0675	1.1120
P02788	TRFL	Lactotransferrin	7.6776	2.2862	1.4302
Q5SQ64	LY66F	Lymphocyte antigen 6 complex locus protein G6f	2.1107	1.3705	1.0059
P54108	CRIS3	Cysteine-rich secretory protein 3	1.7124	1.7709	1.2384
P54578	UBP14	Ubiquitin carboxyl-terminal hydrolase 14	1.6928	0.8851	1.3075
Q7L5Y6	DET1	DET1 homolog	2.8429	0.6266	0.5570
P19971	TYPH	Thymidine phosphorylase	3.9674	3.6955	0.5058
P00488	F13A	Coagulation factor XIII A chain	3.3421	3.2396	0.9832
P06737	PYGL	Glycogen phosphorylase. liver form	8.5721	6.9374	0.5571
P62136	PP1A	Serine/threonine-protein phosphatase PP1-alpha catalytic subunit	1.7576	2.1843	0.9561
P30101	PDIA3	Protein disulfide-isomerase A3	1.7864	0.5549	0.4749
P55786	PSA	Puromycin-sensitive aminopeptidase	2.2783	0.7016	0.6691
P07741	APT	Adenine phosphoribosyltransferase	3.7614	3.9742	0.8305
Q99623	PHB2	Prohibitin-2	1.7628	2.1771	1.6235

Fold-change values in red indicate proteins that were not significantly dysregulated compared with healthy controls (*p* > 0.05). Proteins involved in cellular components organization are shown in clear blue, proteins involved in the cellular response to the stimulus in clear green, components of lysosome in clear pink, extracellular matrix binding and cellular adhesion in orange, proteins involved in constitution and organelles function in purple, secretory granules in dark blue and other proteins involved in transport in dark green.

**Table 6 ijms-22-00226-t006:** Proteins downregulated in ERT-treated versus untreated MPS IVA patients. Proteins considered dysregulated are those with a *p*-value < 0.05 and a fold change (FC) > 1.4.

UniProtCode	UniProt Name	Protein Name	CG	Fold Change Relative to Untreated Group	
				ERT-a	ERT-b
Q96AG4	LRC59	Leucine-rich repeat-containing protein 59	ND	0.5690	0.6684
Q12913	PTPRJ	Receptor-type tyrosine-protein phosphatase eta	ND	0.4456	0.5427
P10153	RNAS2	Non-secretory ribonuclease	ND	0.4025	0.1628
Q06323	PSME1	Proteasome activator complex subunit 1	ND	0.6398	0.6990
P27695	APEX1	DNA-(apurinic or apyrimidinic site) endonuclease	ND	0.7912	0.5368
P02042	HBD	Hemoglobin subunit delta	ND	0.3031	0.5340
O43684	BUB3	Mitotic check point protein BUB3	ND	0.5946	0.4602
Q9Y2Y8	PRG3	Proteoglycan 3	ND	0.5416	0.3759

Fold-change values in red indicate proteins that were not significantly dysregulated because the *p* value was >0.05 (*p* > 0.05). Abbreviations: CG, control group; ERT-a: ERT-a, patients sampled before enzyme infusion; ERT-b, patients sampled 24 h after enzyme infusion; MPS IVA, Mucopolysaccharidosis type IVA; ND, not detected.

**Table 7 ijms-22-00226-t007:** Enzymatic activity of GALNS.

UG	EA nM/h/mg	Before ERT	EA nM/h/mg	After ERT	EA nM/h/mg	Healthy Controls	EA nM/h/mg
1	0.2	ERT-a 1	0.7	ERT-b 1	1.8	CG 1	4.8
2	0.2	ERT-a 2	0.6	ERT-b 2	2.1	CG 2	14.2
3	0.2	ERT-a 3	0.7	ERT-b 3	2.2	CG 3	2.7
4	0.0	ERT-a 4	1.0	ERT-b 4	2.8	CG 4	3.6
5	0.1	ERT-a 5	1.6	ERT-b 5	6.7	CG 5	3.1
6	0.2	-	-	-	-	CG 6	4.8
7	0.1	-	-	-	-	-	-
8	0.2	-	-	-	-	-	-

EA, enzymatic activity; ERT, enzyme replacement therapy; ERT-a, MPS IVA patients sampled before ERT; ERT-b, MPS IVA patients sampled 24 h after ERT; UG, untreated group.

## Data Availability

The data presented in this study are available in this article and this Appendix A.

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
