# Peer review of "Characterization of New Proteomic Biomarker Candidates in Mucopolysaccharidosis Type IVA"

_ijms, 2020, doi:10.3390/ijms22010226_

Round 1
Reviewer 1 Report
Morquio A syndrome (mucopolysaccharidosis type IV A, an autosomal recessive disease caused by mutations in the GALNS gene (encodes GALNS, an enzyme important for the degradation of glycosaminoglycan and glycan structures), is characterized by systemic skeletal dysplasia with incomplete ossification and successive imbalance of growth, including short stature and neck, cervical instability, spinal cord compression, tracheal obstruction, prominent chest, kyphoscoliosis, laxity of joints, hip dysplasia, and knock knees. Since a deficiency of GALNS leads to the accumulation of keratan sulfate and chondroitin-6-sulfate in multiple tissues (mainly in the bone, cartilage, heart valves, and cornea), one of the current therapy is a GALNS enzyme replacement. To better understand the cause of Morquio A syndrome and identify its biomarkers, the authors conducted proteomic analysis using leukocyte samples from 13 patients with MPS IV A (8 untreated and 5 ERT-treated patients) and 6 health control subjects. Although this study is interesting, the sample size is quite small and there is a huge variation in the values of scores tested. In addition, there is the limited information of clinical features of patients with MPS IV A provided in this study and there is a lack of the information of the normal range in a 6-minitutes walk test etc. Since the severity of the disease varies in this study, the authors may add discussion how the severity of the disease was related to the expression level of candidate proteins. Overall, this study is interesting but considered to be at preliminary stage of study or a case report at this point.
Major concerns:
The proteomic analysis was conducted using leukocytes isolated from blood samples. It is unclear how expression of these proteins in leukocytes linked to the phenotypes in bones and cartilages in the patients. Are there any immune abnormalities in patients with MPS IV A? What are the functions of the candidate proteins in leukocytes?
There is a huge variation in the total number of proteins detected through LC-MS/MS even in healthy control and untreated group. Therefore, the statistical analysis should be included in Tables 3-5.
None of the candidate molecules responded a current ERT-treatment (ERT-a versus ERT-b) as shown in Figures 6 and 8. Therefore, it is unclear if candidate proteins are potential biomarkers.
There are significant differences in expression of TRFL and VTNC between UG and ERT groups (Figure 8). Can a long-term treatment improve the clinical features? There is no change in the expression before and after the ERT.
The size of each group is quite small; therefore, it requires comparing the protein expressions between 6 health controls and 13 MPS IV A patients.
Author Response
Review 1
Comments and Suggestions for Authors
Morquio A syndrome (mucopolysaccharidosis type IV A, an autosomal recessive disease caused by mutations in the GALNS gene (encodes GALNS, an enzyme important for the degradation of glycosaminoglycan and glycan structures), is characterized by systemic skeletal dysplasia with incomplete ossification and successive imbalance of growth, including short stature and neck, cervical instability, spinal cord compression, tracheal obstruction, prominent chest, kyphoscoliosis, laxity of joints, hip dysplasia, and knock knees. Since a deficiency of GALNS leads to the accumulation of keratan sulfate and chondroitin-6-sulfate in multiple tissues (mainly in the bone, cartilage, heart valves, and cornea), one of the current therapy is a GALNS enzyme replacement. To better understand the cause of Morquio A syndrome and identify its biomarkers, the authors conducted proteomic analysis using leukocyte samples from 13 patients with MPS IV A (8 untreated and 5 ERT-treated patients) and 6 health control subjects. Although this study is interesting, the sample size is quite small and there is a huge variation in the values of scores tested. In addition, there is the limited information of clinical features of patients with MPS IV A provided in this study and there is a lack of the information of the normal range in a 6-minitutes walk test etc. Since the severity of the disease varies in this study, the authors may add discussion how the severity of the disease was related to the expression level of candidate proteins. Overall, this study is interesting but considered to be at preliminary stage of study or a case report at this point.
Thank you very much for your critical comments. We have revised the manuscript according to your suggestions. We hope you find the revised manuscript acceptable for publication.
Major concerns:
1.The proteomic analysis was conducted using leukocytes isolated from blood samples. It is unclear how expression of these proteins in leukocytes linked to the phenotypes in bones and cartilages in the patients. Are there any immune abnormalities in patients with MPS IV A? What are the functions of the candidate proteins in leukocytes?.
We thank the reviewer for their critical comments. Indeed, analyzing leukocytes is not the same as analyzing chondrocytes in bone, which are the primary target in Morquio A disease. Current ERT does not effectively improve bone lesions, since the infused enzyme fails to fully reach the avascular cartilage tissue. Ideally, this should be tested using cartilage tissue from MPS IVA patients. However, cartilage tissue biopsy is an invasive process. Therefore, we selected leukocytes, which should take up the infused enzyme easily. Using leukocytes, we were able to determine which abnormally expressed proteins can be normalized. Our findings could have important implications for corrective treatment of chondrocytes in future studies, once the treatment that effectively targets bone becomes available.
Regarding immune abnormalities, among the side effects reported by different authors in Morquio A disease, and discussed in the Discussion section of our manuscript (paragraph 5), are those caused by accumulations of substrate in lysosomes. These accumulations generate oxidative stress and rupture of the lysosome membrane, resulting in the release of components into the cytoplasm. These processes ultimately give rise to inflammation and activation of interleukins. In our study, inflammation was observed only in untreated patients, in which prostaglandin pathways are continuously activated.
Primary storage materials (keratan sulfate and chondroitin-6-sulfate) affect various cellular functions, including the immune response via oxidative stress. Lactotransferrin is responsible for regulating oxidative stress within cells and successive inflammatory processes. The functions of lactotransferrin in bone are described in the Discussion section (paragraph 5) and in reference 65. Another candidate protein is Coronin 1A, which is involved in vesicle transport and is a crucial component of the cytoskeleton that is necessary for lysosomal trafficking. In addition, it interacts with cathepsins, specifically cathepsin D in leukocytes. GANAB is a protein found in the endoplasmic reticulum. This protein participates in the generation of proteoglycans related to keratan sulfate. Vitronectin is mainly implicated in cell adhesion.
- There is a huge variation in the total number of proteins detected through LC-MS/MS even in healthy control and untreated group. Therefore, the statistical analysis should be included in Tables 3-5.
The variation in the total number of proteins identified in different samples may be due to the limitations of the techniques used in the proteomic assay. Intersample variability can occur at the level of protein extraction and protein concentration measurement, resulting in differences in the number of proteins identified. Moreover, variability can arise due to the limitations of the LC-MS/MS method used. In determining the total number of proteins, for which a mass search is performed using the DDA method, the aim is to search for all proteins in a given sample. However, in some cases proteins can be masked by the overlapping of multiple proteins in the chromatograph (this can be a problem with healthy controls). This is because the ions enter the mass spectrometer together and only the most numerous ions are identified. To ensure that the technique used is reliable, the proteins in each sample are filtered by the most restrictive FDR (1%). Moreover, despite the risk of potentially missing proteins of interest, in our assay we only select proteins identified in n-1 samples, where n is the total number of samples per group. Data pertaining to the FDR are included in the legend of Table 2.
The quantitative proteomics technique used (SWATH) is based on the creation of a library and a mass search (MRM) of these proteins in individual samples prior to quantification. Only proteins with a 1% FDR were selected for quantification. Of these proteins, only those with 10 peptides and 7 fragments per peptide were extracted and quantified. The proteins selected for quantitative evaluation, as mentioned in the text (first paragraph section 2.1,2), are those with a p-value <0.05 and a fold change value of 1.5 (FC >1.4). These data have been included in Tables 3–5 in accordance with the reviewer’s suggestion.
The total number of proteins identified in all samples using the SWATH method is equal to the number of proteins that appear in the library (690). This is indicated in the abstract and in the discussion of the limitations of the technique.
- None of the candidate molecules responded a current ERT-treatment (ERT-a versus ERT-b) as shown in Figures 6 and 8. Therefore, it is unclear if candidate proteins are potential biomarkers.
These samples were collected before and after treatment, and the after treatment samples were collected 24 h after treatment. We observed a clear tendency towards normalization of the expression of various proteins to levels close to those of healthy controls. The following text has been included at the end of the discussion:
“Another limitation of our study is that normalization of protein expression in leukocytes does not necessarily correlate with normalization in chondrocytes in the avascular region. The enzyme is easily taken up by the leukocytes, restoring the function of dysregulated metabolic pathways in these cells while the infused enzyme circulates in the blood. It is critical to understand whether these proteins are dysregulated in bone and cartilage in MPS IVA patients in order to identify potential diagnostic biomarkers of disease severity. When a bone-penetrating drug becomes available, bone biomarkers will be essential. "
- There are significant differences in expression of TRFL and VTNC between UG and ERT groups (Figure 8). Can a long-term treatment improve the clinical features? There is no change in the expression before and after the ERT.
Response: TRLF and VTNC were found in liver and in bone as described in the discussion section and supported by references 64 and 65. Regarding TRLF, we state the following: “This protein is also implicated in iron metabolism [64] and bone regeneration [65]”. In the same section we point out that “Given their involvement in bone metabolism, we consider TRFL, CORO1A, GANAB and VTNC to be candidate biomarkers of bone impairment in MPS IVA. We postulate that these proteins expressed in leukocytes may also be expressed in bone cells, given that macrophages and osteoclasts originate from the same cell line and may share similar proteins [65]. In a previous proteomic analysis of fibroblasts, we observed upregulation of TRFL in untreated MPS IVA patients and upregulation of GANAB in ERT-treated versus untreated MPS IVA patients [61].”
5) The size of each group is quite small; therefore, it requires comparing the protein expressions between 6 health controls and 13 MPS IVA patients.
We agree with the reviewer’s comment. However, MPS IVA is very rare (1 per 250,000 births in Spain). Therefore, in this pilot study we were unable to achieve larger sample sizes. We will continue to collect and study more cases in future studies.

Reviewer 2 Report
Dear Authors,
congratulations for your work. It is a very interesting paper that face a critical point in the treatment of MPS IV patients.
I agree with all the points and with the unmet needs for these patients.
I only suggest to specify better some abbreviations (as TRLF, VTCN) that may make difficult the reading and to point out better the four potential protein biomarkers.
Author Response
Review 2
Congratulations for your work. It is a very interesting paper that faces a critical point in the treatment of MPS IV patients. I agree with all the points and with the unmet needs for these patients.
1)I only suggest to specify better some abbreviations (as TRLF, VTCN) that may make difficult the reading and to point out better the four potential protein biomarkers.
We thank the reviewer for their suggestion. The names of the proteins have been included in the text in the revised manuscript.
Round 2
Reviewer 1 Report
The authors addressed all the concerns raised by this reviewer.